# The impact of rural e-commerce participation on farmers' entrepreneurial behavior: Evidence based on CFPS data in China

**Haiying Lin[1], Huayuan Wu[1], Haihua Lin[1], Tianqi Zhu[2], Muhammad Umer Arshad[3,4], Haonan Chen[5], Wenlong Li[1] ***

**1** Inner Mongolia University of Finance and Economics, Hohhot, Inner Mongolia, China, **2** Inner Mongolia Open University, Hohhot, Inner Mongolia, China, **3** Department of Crop Sciences, University of Illinois at Urbana-Champaign, Urbana, Illinois, United States of America, **4** Inner Mongolia Honder College of arts and Science, Hohhot, Inner Mongolia, China, **5** School of Economics, Capital University of Economics and Business, Hohhot, China

\* 84599950@qq.com

**Data Availability Statement:** The data underlying the results presented in the study are available from the 2020 China Family Panel Studies

## Abstract

The "Three Rural Issues", encompass challenges related to agriculture, farmer, and rural area, which hold significant importance in driving comprehensive rural revitalization efforts in China. Farmer entrepreneurship, as a crucial means to enhance productivity, create job opportunities, and increase residents' income, has gradually become a key driving force in promoting rural revitalization in the new stage of development in China. With the rapid development of rural e-commerce, farmer entrepreneurship has encountered new opportunities. This study utilizes the 2020 China Family Panel Studies (CFPS) data and employs a structural equation model (SEM) to analyze the direct impact of rural e-commerce participation on farmer entrepreneurial behavior, considering factors such as human capital, social capital, and network infrastructure. This study further explores the indirect effects and mechanisms of e-commerce participation as a mediating variable and analyzes the impact and mechanisms on agricultural entrepreneurship behavior. The findings are as follows: (1) E-commerce participation significantly promotes farmer entrepreneurial behavior; (2) E-commerce participation as a mediating variable has a positive indirect effect on the relationship between social trust, network infrastructure, human capital, and farmer entrepreneurial behavior; (3) E-commerce participation has a significant positive influence on farmer entrepreneurship in the agricultural sector, and farmers with higher levels of network infrastructure and human capital have a higher probability of choosing agricultural entrepreneurship under the influence of e-commerce participation. Finally, this study provides policy recommendations in terms of infrastructure construction, entrepreneurial policy environment, and education level, aiming to optimize the situation of farmer entrepreneurship and contribute to the comprehensive promotion of rural revitalization.Overall, the research in this paper effectively combines theory and empirical evidence to outline the direct and indirect impact mechanisms of rural e-commerce participation on farmers' entrepreneurial behavior and agriculture-related entrepreneurial behavior and to test the effects of their impacts. First, most of the existing literature deals with farmers in individual sample areas, while the sample

database. It is a free and public Chinese database, everyone can download data vie the following link: http://www.isss.pku.edu.cn/cfps/download.

**Funding:** This work was supported by the National Key Research and Development Program of China - Science & Technology Cooperation Project of the Chinese and Russian Governments, entitled "Sustainable Transboundary Nature Management and Green Development Modes in the Context of Emerging Economic Corridors and Biodiversity Conservation Priorities in the South of the Russian Far East and Northeast China" (No. 2023YFE0111300); the National Social Science Fund of China (grant number 23BGL204); the Natural Science Foundation of Inner Mongolia (grant numbers 2021MS07011, 2022MS04001); the Program for Young Talents of Science and Technology in Universities of the Inner Mongolia Autonomous Region (grant numbers NJYT22113, NJYT20B31); and the Inner Mongolia University Direct Scientific Research Business Fee Project (grant numbers NCYWR22020, NCYWR22021).

**Competing interests:** The authors have declared that no competing interests exist.

selected in this paper is farmers in the whole country, which is relatively more generalizable; second, most of the previous studies explore the level of e-commerce in the inter-provincial or county areas, while this paper expands the empirical study of rural e-commerce on the entrepreneurial behavior of farmers and the micro-period of agricultural entrepreneurial behavior, and focuses on the impacts of the e-commerce activities of farmers on their entrepreneurial behavior.

## 1. Introduction

In recent years, the rapid advancement of e-commerce has brought about significant transformations in various sectors of the economy, including the agricultural industry [1]. As digital platforms continue to penetrate rural areas, farmers are increasingly engaging in e-commerce activities as a means to enhance their entrepreneurial endeavors [2]. This paradigm shift has led to a growing interest in understanding the impact of rural e-commerce participation on farmers' entrepreneurial behavior [3]. While several studies have examined the relationship between e-commerce and entrepreneurship [4, 5], there is a paucity of research specifically focusing on the agricultural context, particularly in the Chinese setting.

The 20th National Congress of the Communist Party of China pointed out the need to comprehensively promote rural revitalization and ensure stable and increased agricultural production as well as steady income growth for farmers. The No.1 Central Document for the year 2023 emphasizes the utmost importance of resolving the "Three Rural Issues" as a top priority for the Communist Party of China (CPC) (reference). It aims to promote increased income and employment opportunities for farmers while actively fostering rural entrepreneurial leaders. Therefore, farmer entrepreneurship has gradually become a key means of enhancing productivity, creating employment opportunities, and increasing farmers' income in China's new development stage, serving as an important catalyst for revitalizing rural areas, promoting sustainable agricultural development, and advancing comprehensive rural revitalization [6].

In 2015, the State Council issued the "Opinions on Supporting Rural Migrant Workers and Others to Start Businesses in Their Hometowns," which explicitly stated the need to support the entrepreneurship of rural migrant workers, college graduates, and retired soldiers, and to promote widespread entrepreneurship and innovation to revitalize various industries in rural areas. Subsequently, the government introduced various agricultural subsidies, insurance policies, and financial support policies to encourage rural entrepreneurship. In April 2022, the Director of the Development Planning Department in Ministry of Agriculture and Rural Affairs of the People's Republic of China, Zeng Yande, mentioned that there were over 11 million people engaged in returning to their hometowns and starting businesses across the country, and according to comprehensive calculations, an average entrepreneurial project can provide stable employment for 6 to 7 farmers and flexible employment for 17 individuals [7]. The phenomenon of "one person starting a business and lifting others out of poverty" is emerging, with numerous representative farmer entrepreneurial projects and examples appearing, and the scale of farmer entrepreneurship continues to expand. With the rapid popularization of the internet, rural entrepreneurship has also encountered new opportunities [8].

As of December 2022, the internet penetration rate in China has reached 74.4% high, and the rural internet penetration rate has exceeded 60% [9]. E-commerce has gradually become an important driving force for sustainable economic development in rural areas. The widespread adoption of rural e-commerce plays a crucial and positive role in enhancing the confidence and probability of farmers' entrepreneurship [10–12]. Rural e-commerce not only helps address the issues of lengthy agricultural product distribution channels and information

asymmetry between farmers and consumers, but also improves farmers' profit margins and agricultural product circulation efficiency. It can also reduce farmers' daily living costs [13–15], enhance their human capital, accumulate social capital [16], and promote employment and entrepreneurship among farmers [17]. Since the 18th National Congress of the Communist Party of China, the Central Committee of the Communist Party of China, the State Council, and various ministries have successively launched more than 170 policies [18] and documents to promote the development of rural e-commerce. The implementation of the "Promoting Agricultural Development through Digital Commerce" project has actively promoted the penetration of e-commerce into rural areas and accelerated the construction of e-commerce infrastructure. Particularly, after the Ministry of Commerce launched the "e-commerce in rural areas" initiative, rural e-commerce in China has experienced rapid development. The total rural online retail sales increased from 180 billion yuan in 2014 to 21.7 trillion yuan in 2022, with over 16.32 million rural online businesses and shops, and the number of Taobao villages surpassing 7,000. A large proportion of farmers have found employment and entrepreneurial opportunities through rural e-commerce, leading to increased income levels [19].

In terms of existing literature, since entrepreneurship has become an important driving force for economic growth in various countries, academics have maintained a passion for research on entrepreneurship (Fitz-Koch et al, 2018; Kuratko et al, 2015) [20, 21], which has so far involved a number of disciplinary fields such as psychology, economics, management, law, and so on. Among them, studies related to farmers' entrepreneurship are getting richer and richer, and the research perspectives are endless, mainly including entrepreneurial behavior, entrepreneurial willingness, entrepreneurial field, entrepreneurial performance, entrepreneurial environment and so on. From the view of existing literature, the research on the influencing factors of farmers' entrepreneurial behavior mainly focuses on the following levels: first, the human capital level, such as the level of education [22], the experience of migrant labor, etc. [23, 24]; second, the level of social capital, which mainly includes the social network and social trust, and scholars have shown that empirical studies show that social network and social trust can help to increase the probability of farmers' entrepreneurial activities [25, 26]; Third, the Internet level, exploring the impact of network base on farmers' entrepreneurial behavior is rich in literature, and many studies show that the Internet has a significant and positive impact on farmers' entrepreneurial behavior [27, 28]. Some scholars have also studied the impact of macro-social environmental factors such as financial support, land transfer and other policy support factors on farmers' entrepreneurial behavior [29–31]. With the gradual sinking of information technology in rural areas, the opportunities brought by rural e-commerce for farmers' entrepreneurship have begun to trigger discussions among scholars, and the current research on the impact of rural e-commerce on farmers' entrepreneurial behavior is mainly from two perspectives, one is to study the direct or indirect impact of rural e-commerce level on farmers' entrepreneurship, and the majority of scholars have shown that the level of rural e-commerce has a significant positive impact [32, 33]. The second is to analyze farmers' e-commerce entrepreneurial behavior, i.e., the entrepreneurial behavior of economic and trade activities through electronic means, such as Taobao stores, microbusinesses, and live streaming with goods, etc. E-commerce is conducive to breaking the differences in entrepreneurial opportunities caused by the differences in social capital, and to breaking the limitations imposed by the traditional social capital on farmers' entrepreneurship, which indirectly improves the confidence of farmers' entrepreneurship, and promotes farmers' entrepreneurship [34]; some scholars have also carried out different regional farmers' e-commerce entrepreneurial behavior has been studied in depth, such as Jiangxi Province [8], Guizhou Province and Chongqing Municipality [35], Jiangsu Province and so on [36, 37]. Rural e-commerce participation is a core branch of rural e-commerce, which is also known as farmers' e-

commerce adoption, farmers' participation in e-commerce, and so on. For rural e-commerce participation there is no clearer definition, and the existing literature on the connotation of rural e-commerce participation mainly involves the following dimensions: first, farmers' online shopping behavior in the common sense, Chinese scholar Han Feiyan pointed out through empirical research that online shopping improves farmers' knowledge of e-commerce, stimulates their consumption vitality, and breaks down the market information barriers [38]; the second concept is in the further development on the basis of the first, i.e., farmers enter the e-commerce market as sellers, participate in market sales, and utilize rural e-commerce to improve their income. Ma Jingtao defines rural e-commerce as a business activity centered on the trading of agricultural products under the influence of the Internet environment and with the help of various mobile communication devices. The rural e-commerce participation in this paper focuses on the first concept. Currently, there are still few studies in this area, and the existing literature focuses mainly on the willingness to participate in rural e-commerce and its influencing factors, and the impact of rural e-commerce participation on farmers' income. However, considering the rapid development of rural e-commerce and its important role in farmers' entrepreneurship and rural economic development, there is still a lot of room for expansion of research in this area. First, Hao Jinlei and Xing Xiangyang (2016) studied farmers' willingness to participate in e-commerce at the levels of literacy, government support, and commercial bank coverage, and further pointed out the importance of this behavior for the development of rural e-commerce [39]; Guo Jinlong et al. (2023) empirically analyzed the influencing factors of farmers' e-commerce participation behavior from the perspective of social interactions, arguing that social interactions positively influences farmers' e-commerce participation behavior by changing farmers' information cognitive norms and social norms [40]; Lin Haiying et al. (2020) conducted an empirical study on the e-commerce participation willingness of farmers and herdsmen in some impoverished areas of Inner Mongolia and the factors influencing them, and found that factors such as infrastructure, personal traits, social networks, and resource endowment significantly affect farmers' and herdsmen's e-commerce participation willingness [13]. Secondly, Cao et al. (2021) explored the impact of e-commerce participation on rural gender income gap based on the theory of household specialization division of labor and gender comparative advantage theory [41]. Yu Hao et al. (2021) focused on the research data of 303 farming households in Shaanxi Province and applied the Fields income decomposition method to explore the direction and effect of e-commerce participation on the income gap of farming households [42]. Furthermore, the role of agriculture and farmers is particularly important in the context of the current global food security situation. It has been pointed out that, in addition to economic aspects, the impact of agricultural production on the environment, landscape, and land use is more prominent than that of other economic sectors (Britz et al., 2012) [43], and it is crucial to understand how farmer-agricultural entrepreneurs acquire entrepreneurial capabilities (Pindado et al, 2017; Seuneke et al, 2013) [44, 45]. Agricultural entrepreneurial behavior, on the other hand, is an important tool to improve farmers' income and promote agricultural diversification, and the current international research literature on agricultural entrepreneurial behavior mainly includes the role of agricultural entrepreneurial behavior, and the influencing factors of agricultural entrepreneurial behavior. Some scholars have studied farmers' entrepreneurial behavior in agriculture-related tourism, pointing out that farmers' entrepreneurial behavior in agriculture-related tourism is an effective way to strengthen the construction of rural spiritual civilization and alleviate the pressure of urban and rural employment, and encouraging more farmers to carry out entrepreneurial activities related to agriculture [46]. Some scholars have deeply explored the impact of farmers' entrepreneurship on the ecological environment of farmland and pastureland, pointing out that agricultural entrepreneurship provides opportunities for

environmentally sound use of resources, which can serve as an important remedy for environmental challenges, and is mainly affected by market opportunities, infrastructure, especially network facilities, educational level, social capital, etc. [47, 48].George Saridakis et al. (2021) through the method of empirical analysis pointed out that agricultural entrepreneurship as a business activity can be effective in improving economic well-being [49].While China's research in related fields started late, the earliest one to conduct a more in-depth exploration of agricultural entrepreneurial behavior was Yu Ning (2013) in China, whose dissection of the importance of agricultural entrepreneurial behavior and the related influence mechanism laid a solid foundation for the research in the field of agricultural entrepreneurial behavior in China [50]. In recent years, some scholars in China have begun to pay attention to how to cultivate high-quality talents for agricultural entrepreneurship in the new era, and some scholars focus on agriculture-related colleges and universities and college students returning to their hometowns to engage in agricultural entrepreneurship (Hao Zhenping et al., 2023; Zhao Fangfang et al., 2023) [51, 52], and most of them use theoretical analysis and lack of empirical data to support it; there are also individual scholars who have used the method of empirical analysis of a certain region's farmers' agricultural entrepreneurial behavior has been explored [53]. However, in general, there is a lack of articles in the new development stage that use empirical methods to conduct an in-depth exploration of farmers' agricultural entrepreneurial behavior and the influencing factors behind it on a national scale.

In summary, the current research on farmers' entrepreneurial behavior has been relatively rich, first, scholars have analyzed the influence factors of rural e-commerce on farmers' entrepreneurial behavior from different levels and regions, but most of the scholars have only studied the influence of a single level on farmers' entrepreneurial behavior, such as social capital or network base, and very few literatures have explored these factors in a comprehensive way; second, there is a lot of literature exploring the influence of rural Secondly, there is also a lot of literature exploring the impact of rural e-commerce on farmers' entrepreneurial behavior, but most of the existing studies use probit model, binary logistic model and other methods to study the impact of rural e-commerce on farmers' entrepreneurial behavior from the perspective of e-commerce level of the provinces and counties, and there is rarely any literature that analyzes the impact of individual farmer's participation in e-commerce on his or her entrepreneurial behavior by using structural equation modeling. Again, there are fewer studies on rural e-commerce participation, and most of them are based on empirical studies in a specific region, which lacks a certain universality;In addition, although existing studies mention the impact of rural e-commerce participation on farmers' income, there is a lack of empirical studies related to the impact of rural e-commerce participation on farmers' entrepreneurial behavior and agricultural entrepreneurial behavior. Overall, at the level of theoretical contribution, this paper may make some contributions to the research field of farmers' entrepreneurship and rural e-commerce in the following aspects: (1) Most of the existing literature involves farmers in individual sample areas, whereas this paper selects a sample object of farmers in the whole country, which is relatively more generalized. (2) Most of the previous studies explore the level of e-commerce in inter-provincial or county areas, while this paper expands the empirical study of rural e-commerce on farmers' entrepreneurial behavior and agricultural entrepreneurial behavior from a micro point of view, and the study focuses on the impact of individual farmers' e-commerce activities on farmers' entrepreneurship. This study focuses on the impact of individual farmers' e-commerce activities on farmers' entrepreneurship, combines the existing authoritative literature and the real-world background, analyzes the theoretical mechanism of the impact of rural e-commerce participation on farmers' entrepreneurial behavior and agricultural entrepreneurial behavior under the role of multiple factors, and explores the key influencing factors, which is conducive to supplementing and perfecting the theoretical system of farmers'

entrepreneurial behavior, especially agricultural entrepreneurial behavior. (3) The importance of rural e-commerce participation in agricultural entrepreneurial behavior is rarely mentioned in the existing literature, especially in developing countries such as China, which has a large number of mountainous areas where mechanization of agricultural production is difficult. E-commerce is a bridge between Chinese farmers and external producers, rural e-commerce participation in farmers' agricultural entrepreneurial behavior is the most direct impact is that it can make the farmers of agricultural products on the network sales of low-cost, low-threshold, and the possibility of understanding the market mechanism is greater, for the long term to maintain the livelihood of the farmers in agricultural production, rural e-commerce is the importance of agricultural entrepreneurship is not negligible, but the literature does not have an in-depth discussion of this phenomenon. There is no literature to explore this phenomenon in depth, and this paper enriches the theoretical research in this field to a certain extent. On the practical level, this study utilizes research data to provide an in-depth discussion on the importance of farmers' participation in e-commerce and entrepreneurial activities, and tests the intrinsic factors affecting farmers' entrepreneurship in the empirical analysis, such as human capital and social capital, which can provide theoretical references for the farmers who hope to achieve income growth and improve their living standards through entrepreneurship, so that they can have a certain way of thinking about the preparation for their entrepreneurship, and improve the farmers' It can provide theoretical references for farmers who wish to increase income and improve living standards through entrepreneurship, so that they can have certain ideas to prepare for entrepreneurship and improve their motivation and self-confidence. In addition, the research results of this paper can help the relevant government departments to formulate targeted policies to create a favorable entrepreneurial environment for the promotion of farmers' entrepreneurship and agricultural development, and provide a certain micro reference basis for improving the level of grassroots governance.

Therefore, this paper aims to address several key questions. Firstly, we utilized the 2020 CFPS data, supplemented by data from 2018, and employs a SEM model to analyze the significant impact of farmers' participation in rural e-commerce on their entrepreneurial behavior and involvement in agricultural entrepreneurship, taking into account various factors such as social capital, network infrastructure, and human capital. Compared to general regression analysis models, the SEM research method allows for the consideration of the correlation between multiple variables and accommodates the presence of multiple dependent variables and measurement errors. It enables the simultaneous observation of both the direct and indirect effects of e-commerce participation on farmers' entrepreneurial behavior and involvement in agricultural entrepreneurship. The CFPS database covers individual survey data from 25 provinces nationwide, providing a broader sample representation and thus enhancing the generalizability of the empirical results. Secondly, we explored the indirect effects and mechanisms of e-commerce participation as a mediating variable. Thirdly, based on the empirical analysis, we have proposed some policy recommendations. The aim is to provide effective means for promoting farmer entrepreneurship through rural e-commerce and to offer micro-level reference for advancing high-quality development of the agricultural economy and facilitating comprehensive rural revitalization in China.

## 2. Theoretical analysis and research hypotheses

### 2.1 Direct mechanisms of rural e-commerce impact on farmers' entrepreneurial behavior

For farmers, rural e-commerce participation is not only an emerging way of shopping, but also a means of obtaining information, accumulating entrepreneurial experience, and mobilizing

entrepreneurial capital. First, rural e-commerce participation can help farmers more easily understand market conditions, explore business opportunities, and acquire entrepreneurial knowledge and skills. In the process, they can improve their ability to collect and utilize information, thus laying a solid foundation for entrepreneurship [13]. Second, rural e-commerce participation allows farmers to be exposed to a broader market and observe and learn more business models and business strategies. For example, they can accumulate business experience by comparing the price and quality of goods on different e-commerce platforms. Third, rural e-commerce participation can provide more diversified and affordable goods, effectively reducing farmers' cost of living, helping them save money and accumulate start-up capital.Particularly in some less developed rural areas, there are fewer physical stores and a limited choice of goods due to geographical and economic constraints. Rural e-commerce participation, on the other hand, breaks this limitation and enables farmers to enjoy the same shopping experience and services as urban residents. Not only that, rural e-commerce participation is also conducive to increasing farmers' awareness of market participation. In the traditional agricultural business model, farmers are often only producers, their understanding of the market and participation is limited. However, rural e-commerce participation breaks this limitation, so that farmers can directly contact the market and understand consumer demand and behavior, thus improving their market sensitivity and participation awareness.In addition, e-commerce online transactions are based on credit evaluation and systematic payments, and farmers' understanding of such mechanisms can promote a more trusting attitude towards market mechanisms, enable a wider range of economic transactions, and contribute to farmers' better participation in market transactions, reduce their perceived entrepreneurial risk, and increase their entrepreneurial likelihood [32]. Finally, numerous studies related to farmers' entrepreneurial behavior point out that prior experience has an important impact on entrepreneurial behavior [54], i.e., farmers' entrepreneurial behavior is influenced by their acquired social experience. E-commerce participation is also a type of prior experience that can make it more likely for farmers to understand the low cost, low threshold, and market mechanism of selling agricultural products online, and to accumulate prior experience in e-commerce entrepreneurship, which can contribute to their entrepreneurial behavior.

Based on this, the following hypothesis is proposed.

H1: E-commerce participation promotes farmers' entrepreneurial behavior by enhancing their information gathering ability, accumulating entrepreneurial capital, and acquiring prior experience in e-commerce entrepreneurship.

## 2.2 E-commerce as a mediator for farmers' entrepreneurial behavior

Literature review shows that there are several important factors influencing farmers' entrepreneurial behavior. This study focuses on three factors that are related to both e-commerce participation and farmers' entrepreneurial behavior: human capital, social capital, and network infrastructure. The study argues that e-commerce participation, as a mediating variable, can have an indirect impact on the relationship between these factors and farmers' entrepreneurial behavior.

**2.2.1 Rural e-commerce's impact on human capital -farmers' entrepreneurial behavior.** In this study, farmers' human capital primarily includes their level of education and work experience. The level of education has a significant positive impact on their ability to identify entrepreneurial opportunities [55]. Farmers with higher levels of education are more likely to utilize various platforms to gather and understand entrepreneurial information and

skills [35]. Work experience mainly refers to farmers' experience in working outside their hometown. The accumulated work experience and social exposure gained from working outside the hometown contribute to the development of farmers' individual capabilities and promote entrepreneurship [36]. Firstly, farmers with higher levels of education not only possess higher levels of human capital but also have a stronger ability to accept and learn new things. They are more likely to have a positive risk attitude [56] and are more capable of acquiring the skills required for e-commerce participation, leading them to engage in e-commerce activities different from traditional shopping methods. Secondly, farmers with extensive work experience also exhibit a greater acceptance of new things and are more likely to participate in e-commerce activities. Additionally, under China's dual rural-urban household registration system, discrimination exists against individuals who have no household registration permit in terms of wage determination, social security, and contract guarantees in most urban employment settings [57]. Such unfavorable employment conditions greatly discourage farmers from seeking employment opportunities in urban areas. For entrepreneurial farmers, education accumulates their human capital, while e-commerce participation accumulates their prior experience. Combined with the negative impact of limited employment opportunities in urban areas, they are more likely to identify entrepreneurial opportunities and choose entrepreneurship instead of seeking employment elsewhere [58, 59]. Therefore, this study suggests that farmers with higher levels of education and more extensive work experience are more inclined to choose entrepreneurship under the influence of e-commerce participation.

Accordingly, we proposed Hypothesis H2: E-commerce participation mediates the positive relationship between human capital and farmers' entrepreneurial behavior.

**2.2.2 E-commerce's Impact on social capital-farmers' entrepreneurial behavior.** Social capital refers to the social resources that farmers possess to sustain their livelihoods, pursue their own development, and cope with risky shocks, and social networks and social trust are two of the core elements for measuring social capital [60, 61]. Social capital can accelerate the accumulation of entrepreneurial resources and provide some psychological support for entrepreneurship, which can have a significant impact on an individual's decision to engage in entrepreneurship [62, 63]. Among them, social network support helps to increase the probability of farmers engaging in entrepreneurial activities [60]. In rural areas of China, where there is often a strong sense of interpersonal relationships, farmers who already have entrepreneurial intentions are more likely to seek entrepreneurial funding from relatives and friends in their relational networks [48]. Social trust, in turn, helps to fill the gaps in the formal system, reduces transaction costs and entrepreneurial risks, and facilitates the collection of entrepreneurial information and knowledge, thereby increasing the probability of farmers' entrepreneurship [64]. E-commerce transactions are based on credit assessment and secure payment systems. Farmers who have some knowledge of this mechanism are more likely to trust strangers and thus develop a more trusting attitude towards market mechanisms. This wider range of economic transactions, especially beyond the boundaries of kinship and geographical trust, gradually builds a higher level of broad social trust. This, in turn, helps farmers to better participate in market transactions, reduces their perceived entrepreneurial risk, and increases their likelihood of starting a business. Higher levels of social trust can trigger the "herd effect", whereby farmers' behavioral decisions are influenced by the behavior of the surrounding groups, such as when farmers consider whether to engage in e-commerce participation, they tend to pay close attention to the experiences and feelings of those who have already adopted them [40], which in turn promotes their e-commerce participation behavior, and further facilitates entrepreneurial information collection and knowledge acquisition, thus increasing the probability of farmers' entrepreneurship.In this paper, it is argued that e-commerce participation mainly affects the relationship between social trust in social capital and farmers'

entrepreneurial behavior, but social networks are also included as one of the latent variables in consideration of the rigor and rationality of the model design.

Based on this, we have proposed following hypotheses:

H3: E-commerce participation mediates the positive relationship between social trust and farmers' entrepreneurial behavior.

**2.2.3 E-commerce's impact on network infrastructure-farmers' entrepreneurial behavior.** The Internet has a significant and positive impact on farmers' entrepreneurship [65–67]. And farmers with more experience in Internet use are more likely to engage in promoting their entrepreneurial behavior through e-commerce. First, farmers with correspondingly better familiarity with and trust in online platforms are better able to understand and use relevant e-commerce platforms, such as e-commerce platforms like Pinduoduo, Jingdong, Taobao, and social platforms like WeChat and Jieyin. These platforms can provide space for farmers to display, sell and market their products. At the same time, farmers who are experienced in Internet use have more trust in online platforms, and they are more willing to trade and sell on the Internet, which undoubtedly provides possibilities for their entrepreneurial behavior. Second, farmers who are more skilled in Internet use are relatively more likely to use the Internet to learn about the advantages of e-commerce participation and keenly identify entrepreneurial opportunities in online sales. The Internet provides farmers with a wealth of information resources, and they can learn about the advantages of e-commerce, such as open markets, convenient transactions, and flexible hours, through query searches, browsing web pages, and watching videos. At the same time, they can also use social networks, forums and other platforms to identify and grasp entrepreneurial opportunities in online sales. All these advantages and opportunities may stimulate farmers' entrepreneurial desire and prompt them to take entrepreneurial actions. Finally, the Internet provides rich learning resources. Farmers can learn the basics and skills of online sales, such as e-commerce operations and product promotion, through online courses and video tutorials. Not only that, the Internet connects global information, so farmers can get in touch with the latest sales concepts and methods, compare and learn online sales skills through the Internet, thus ultimately motivating their entrepreneurial behavior.

Accordingly, we have proposed the fourth hypothesis:

H4: E-commerce participation mediates the positive relationship between network infrastructure and farmers' entrepreneurial behavior.

## 3. Data and model

### 3.1 Data

The data used in this study is obtained from the CFPS database. CFPS is a nationwide longitudinal survey conducted by the Institute of Social Science Survey at Peking University. It covers individual-level survey data from 25 provinces in China and provides a representative sample that reflects the development and changes in Chinese society and rural economy. For the empirical analysis in this study, the 2020 CFPS data [68] is primarily used. Considering the stability of social networks and the availability of data, referring to Zhou and his colleagues' method [69] the 2018 survey data is used for the selection of social network observation variables, and the individual-level data from the 2020 CFPS is matched with the household economic data from the 2018 CFPS using the unique personal identifier "pid" in the CFPS

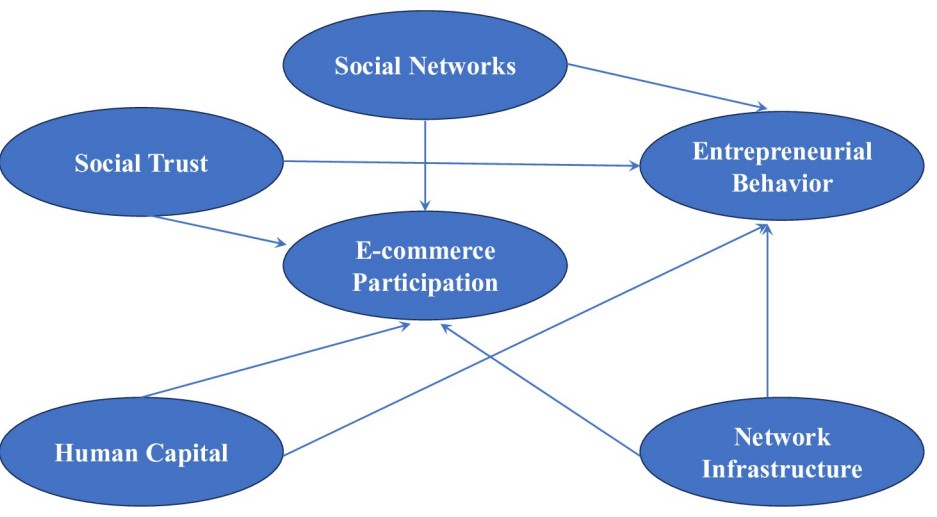

**Fig 1. Initial conceptual model.**

database [70]. Missing values for key variables are excluded, resulting in a final sample size of 4057 valid observations.

## 3.2 Methodology

Given that our study relies on analyzing secondary data and doesn't involve direct interaction with human participants, we didn't seek institutional review board (IRB) approval, as it isn't applicable to our research. Based on the research hypotheses, a conceptual model is constructed to examine the relationships among six latent variables: entrepreneurial behavior, e-commerce participation, social networks, social trust, human capital, and network infrastructure. The model is depicted in Fig 1.

## 3.3 Variable selection and statistic descriptions

This study focuses on rural farmers and comprehensively analyzes the direct impact of rural e-commerce participation on entrepreneurial behavior, taking into account factors such as human capital, social capital, and network infrastructure. It also explores the indirect effects and mechanisms of e-commerce participation as a mediating variable, further examining the effects and mechanisms on agricultural entrepreneurship. The sample of farmers is selected based on the "rural/urban attribute of permanent residence" indicator from the 2020 and 2018 CFPS databases. Taking into account practical considerations and referencing previous literature [54, 71–75], a total of 10 observed variables are selected from the individual survey questionnaire to reflect the five latent variables of entrepreneurial behavior, e-commerce participation, human capital, network infrastructure, and social networks.

**3.3.1 Entrepreneurial behavior.** Constructed as a binary variable to measure the entrepreneurial behavior of farmers based on the CFPS 2020 questionnaire item "What is your current main job/most recent job type?". If the response is "own agricultural production and operation/private enterprise/individual business/self-employment", the variable is assigned a value of 1; otherwise, it is assigned a value of 0.

**3.3.2 E-commerce participation.** Constructed based on the CFPS 2020 question "Have you made online purchases in the past week?" to measure the e-commerce participation of

farmers. If the response is "yes", the variable is assigned a value of 1; otherwise, it is assigned a value of 0.

**3.3.3 Social networks.**　In rural China, maintaining social connections through gift-giving is an important tradition. The variable is constructed based on the CFPS 2018 question "Including goods and cash, how much did your household spend on gifts in the past 12 months?" using the reported value of "gift expenditures (in yuan/year)".

**3.3.4 Social trust.**　Given that social trust cannot be directly observed and has rich connotations, this latent variable is measured using three observed variables. Based on the CFPS 2020 questionnaire, the item "In general, do you think most people can be trusted or is it better to be cautious when dealing with others?" is used. If the response is "most people can be trusted", the observed variable "liking to trust or doubt others" is assigned a value of 1; if the response is "better to be cautious when dealing with others", it is assigned a value of 0. Additionally, the variable "trust in strangers" is constructed based on the respondent's rating on a scale from 1 to 10, indicating the level of trust in strangers. Similarly, the variable "trust in local government" is constructed based on the respondent's rating on a scale from 1 to 10, indicating the level of trust in the local government.

**3.3.5 Network infrastructure.**　Mainly refers to the respondent's internet usage. The observed variable "importance of the internet as an information channel" is constructed based on the CFPS 2020 question "Rate the importance of the internet as an information channel" on a scale from 0 to 5. The observed variable "daily duration of mobile internet usage" is constructed based on the CFPS 2020 question "How many minutes do you spend on average using the internet on your mobile device?"

**3.3.6 Human capital.**　Mainly considers the respondent's education level and work experience. The observed variable "education level" is constructed based on the CFPS 2020 question "What is the highest level of education you have completed (graduated from)?" with values assigned as follows: "0 = illiterate/semi-literate, 3 = primary school, 4 = junior high school, 5 = high school/vocational school/technical school, 6 = college, 7 = bachelor's degree, 8 = master's degree, 9 = doctorate". The observed variable "full-time work experience" is constructed based on the CFPS 2020 question "Have you ever had full-time work experience?" with a value of 1 assigned for "yes" and 0 for "no".

Based on the initial conceptual model diagram (Fig 1), a SEM diagram is constructed (Fig 2), and descriptive statistics are performed. The results are presented in Table 1.

Table 1 shows among the surveyed individuals, approximately 39% of farmers have engaged in entrepreneurship, and around 57% of farmers have a high level of e-commerce participation. The observable variable "gift expenditures" under the category of "social networks" has a mean value of 4092.96 and a standard deviation of 5317.874, indicating significant variation in gift expenditures among farmers. Among the observed variables in the "social trust" category, the mean value for "preference for trusting or doubting others" is 0.59, indicating that the level of social trust among surveyed farmers is generally moderate to slightly higher. The mean value of "trust in strangers" is 2.49, with a mode of 0, suggesting that most farmers have a low level of trust in strangers. However, the mean value of "trust in local government" is 5.66, with a standard deviation of 2.523 and a mode of 5, indicating that farmers generally hold a moderate to slightly higher level of trust in the local government. Regarding "network infrastructure," the mean value for "daily duration of mobile internet usage" is 158.24, with a mode of 120 minutes and a standard deviation of 156.151. This suggests that the daily internet usage duration of farmers varies significantly, with the majority spending around two hours online. The mean value for the importance of the internet as an information channel is 4.11, with a standard deviation of 1.067 and a mode of 5. This indicates that the internet is considered an important source of information for the majority of surveyed farmers. For the "human capital" category,

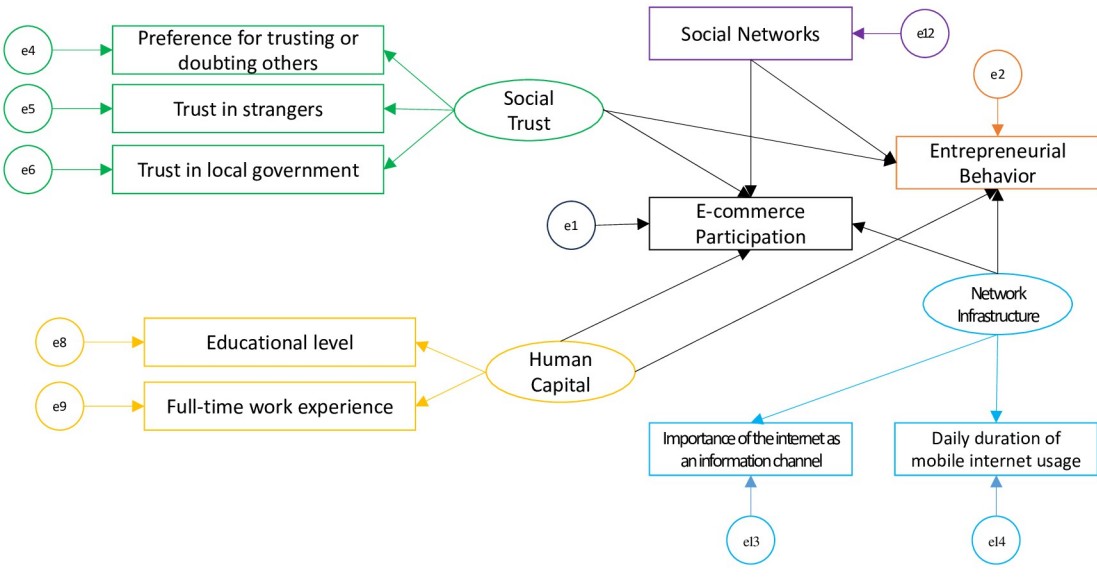

**Fig 2. Initial conceptual SEM model structure.**

the mean value of "education level" is 4.19, with a mode of 4, suggesting that the education level of most sampled farmers is at the junior high school level. Overall, the education level is relatively low. The mean value for "full-time work experience" is 0.53, indicating that 53% of sampled farmers have had full-time work experience.

# 4. Empirical analysis

## 4.1 Reliability and validity testing

To ensure the reliability and validity of the data, reliability and validity testing was conducted. To standardize the data and unify the measurement scale, the data was normalized using

**Table 1. Variable definitions and statistic descriptions.**

| Latent Variable | Observable Variable | Description | Min | Max | Mean | Standard Deviation | Mode |
|---|---|---|---|---|---|---|---|
| Entrepreneurial Behavior (EB) | Whether engaged in entrepreneurship | 1 = yes, 0 = no | 0 | 1 | 0.42 | 0.493 | 0 |
| E-commerce Participation (EP) | Online shopping in the past week | 1 = yes, 0 = no | 0 | 1 | 0.55 | 0.498 | 1 |
| Social Networks (SN) | Gift expenditures | Total amount spent on gifts in a year | 0 | 50000 | 4092.96 | 5317.84 | 2000 |
| Social Trust (ST) | Preference for trusting or doubting others | 0 = prefer trust, 1 = prefer doubt | 0 | 1 | 0.59 | 0.493 | 0 |
| | Trust in strangers | Rating from 0 to 10 | 0 | 10 | 2.47 | 2.154 | 0 |
| | Trust in local government | Rating from 0 to 10 | 0 | 10 | 5.66 | 2.523 | 5 |
| Network Infrastructure (NF) | Daily duration of mobile internet usage | Minutes of mobile internet usage per day | 0 | 1200 | 158.24 | 156.15 | 120 |
| | Importance of the internet as an information channel | Rating from 0 to 5 | 1 | 5 | 4.11 | 1.067 | 5 |
| Human Capital (HC) | Education level | 0 = illiterate/semi-literate; 3 = primary school; 4 = junior high school; 5 = high school/vocational school/technical school; 6 = college; 7 = bachelor's degree; 8 = master's degree; 9 = doctorate | 0 | 8 | 4.19 | 1.489 | 4 |
| | Full-time work experience | 1 = yes, 0 = no | 0 | 1 | 0.53 | 0.5 | 1 |

Z-Score before conducting reliability and validity testing, which improved the comparability of the data. The Cronbach's alpha value for the sample is 0.642, and the Cronbach's alpha values for each observed variable are all above 0.6, indicating acceptable questionnaire reliability. The overall KMO (Kaiser-Meyer-Olkin) test coefficient for the sample population is 0.797, and the Bartlett's sphericity test is significant (P = 0.000). The factor loadings of the six latent variables on their respective measurement items are significant, consistent with the proposed hypotheses of the model, indicating that the questionnaire data is suitable for factor analysis and has good validity.

## 4.2 Fitness testing

SEM model is a statistical method based on the covariance matrix of variables to analyze relationships between variables. It can specify a latent variable model to estimate the relationships between latent structures and observed variables, as well as analyze the relationships between multiple independent variables and multiple dependent variables. In this study, a structural equation model was constructed to examine the influence of rural e-commerce participation on farmers' entrepreneurial behavior, incorporating six latent variables: entrepreneurial behavior, e-commerce participation, social network, social trust, network infrastructure, and human capital, along with their corresponding observed variables. The fitness of the structural equation model need to be assessed through a series of fit indices, including absolute fit indices: AGFI, GFI, RMSEA; incremental fit indices: CFI, NFI, IFI; and parsimonious fit index: $\chi^2/df$. AGFI is expected to be greater than 0.85, while GFI, CFI, NFI, and IFI should be greater than 0.9, with values closer to 1 indicating better fit. A smaller RMSEA value indicates better fit, typically below 0.08 for larger samples and below 0.05 for excellent fit. A smaller $\chi^2/df$ value indicates better fit, with values between 2 and 5 considered good fit. The data, which passed reliability and validity testing, were imported into AMOS 24.0 software for model fit testing, and the fit indices are presented in the following table. From Table 2, it can be observed that the overall fit of the hypothesized model is acceptable, as $\chi^2/df$, AGFI, GFI, and RMSEA have met the requirements for model fit indices. However, CFI, NFI, and IFI did not meet the standards, and the values of $\chi^2/df$ and RMSEA are relatively high, indicating that further modifications are needed to improve the model.

## 4.3 Model modification

To improve the model fitness and achieve better results, it was necessary to make adjustments to the initial model. This study primarily employed model modification methods by observing the modification indices (MI) in the model estimation results and following the principle of releasing one parameter at a time for model modification. Based on the results from AMOS 24.0 software (Table 3), the model was modified.

At first step we run the model and result showed the highest MI value between the residual items of "duration of mobile internet usage" and "educational level," which reached 188.899.

**Table 2. Initial model fit results.**

| Fit Indices | Absolute Fit Index | | | Incremental Fit Index | | | Parsimonious Fit Index |
|---|---|---|---|---|---|---|---|
| | AGFI | GFI | RMSEA | CFI | NFI | IFI | $\chi^2/df$ |
| Fit Standards | >0.85 | >0.9 | <0.08 | >0.9 | >0.9 | >0.9 | Smaller is better |
| Results | 0.935 | 0.966 | 0.077 | 0.761 | 0.755 | 0.762 | 24.904 |
| Fit Assessment | Good | Good | Reasonable | Poor | Poor | Poor | Reasonable |

**Table 3. Model results modified based on MI indices.**

| Modification | Added Path | MI |
|---|---|---|
| First Modification | e8 ↔ e14 | 188.899 |
| Second Modification | e8 ↔ e13 | 105.430 |
| Third Modification | e5 ↔ e8 | 95.931 |
| Fourth Modification | e14 ↔ e9 | 93.081 |
| Fifth Modification | e13 ↔ e6 | 81.815 |
| Sixth Modification | e8 ↔ e4 | 74.023 |
| Seventh Modification | e5 ↔ e9 | 56.852 |
| Eighth Modification | e9 ↔ e13 | 47.024 |

Note: e4 represents the residual item for "preference for trusting or doubting others"; e5 represents the residual item for "trust in strangers"; e6 represents the residual item for "trust in local government"; e8 represents the residual item for "educational level"; e9 represents the residual item for "full-time work experience"; e13 represents the residual item for "importance of the internet as an information channel"; e14 represents the residual item for "daily duration of mobile device internet usage".

Considering the questionnaire data and relevant literature, the educational level influences farmers' learning and cognitive abilities. Farmers with higher education levels are more likely to benefit from internet usage. Therefore, a correlation path was added between the residual items e8 and e14 to improve the model fit. After re-estimating the model, it was observed that the residual items e8 and e13 between "educational level" and "importance of the internet as an information channel" had the highest MI value of 105.430. Similarly, farmers with higher education levels are more likely to benefit from the information dividends brought by the internet. Thus, the correlation paths between e8 and e13 were added.

Further sequential estimation of the model revealed that the residual items with the highest MI value were between "trust in strangers" and "educational level." Literature suggests that educational level is related to the allocation of educational resources, and whether educational resources are fairly and effectively distributed can influence farmers' trust in the general trust network of strangers. Higher education levels are associated with higher levels of general trust [76]. Therefore, a correlation path was added between the residual items e5 and e8 to improve the model fit. Upon re-estimating the model, it was observed that the highest MI value was between the residual items of "duration of mobile internet usage" and "full-time work experience," reaching 93.081. Considering practical perspectives, many job postings are published on the internet, and farmers actively utilize the internet to find employment information. Hence, correlation paths were added between the residual items e14 and e9. Another significant MI value was observed between the residual items of "importance of the internet as an information channel" and "trust in local government," indicating that the widespread use of the internet expands the traditional limitations of information acquisition. It allows farmers to receive more timely policy information from the government, broadening channels for expressing opinions and influencing farmers' trust in the government [77]. Thus, a correlation path was added between the residual items e13 and e6.

At second step we run the model again and results showed a relatively high MI value between the residual items of "educational level" and "trust or suspicion of others," reaching 74.023. Numerous studies have indicated that individuals with more resources are more likely to trust others because resource conditions affect their "disaster line" and vulnerability. Education can improve farmers' resource holding status and promote individual trust [78]. Therefore, a correlation path was added between the residual items e8 and e4. The next significant

**Table 4. Fitness results of the modified model.**

|  | Absolute Fitness Index | | | Incremental Fitness Index | | | Parsimonious Fitness Index |
|---|---|---|---|---|---|---|---|
| Fit Indices | AGFI | GFI | RMSEA | CFI | NFI | IFI | $\chi^2/df$ |
| Fit Standards | >0.85 | >0.9 | <0.08 | >0.9 | >0.9 | >0.9 | Smaller is better |
| Results | 0.984 | 0.994 | 0.035 | 0.963 | 0.956 | 0.963 | 6.106 |
| Fit Assessment | Good | Good | Good | Good | Good | Good | Good |

Note: The mentioned variables (e4, e5, e6, e8, e9, e13, e14) refer to residual variables, such as preference for trusting or doubting others " (e4), "trust in strangers" (e5), "trust in local government" (e6), "education level" (e8), "full-time work experience" (e9), "importance of the Internet as an information channel" (e13), and "daily duration of mobile device internet use for business" (e14).

MI values were observed between the residual items of "trust in strangers" and "full-time work experience" and between "full-time work experience" and "importance of the internet as an information channel," with values of 56.852 and 47.024, respectively. Considering the questionnaire data and actual situations, trust in strangers and previous full-time work experience may be related. Higher trust in strangers among farmers when seeking employment opportunities can lead to lower contract costs and higher job efficiency. Similarly, job quality can also influence farmers' trust in strangers [79]. Therefore, a correlation path was added between the residual items e5 and e9 to reduce the chi-square value. The internet can effectively expand farmers' channels for obtaining employment information, allowing them to find suitable jobs more efficiently. Thus, a correlation path was constructed between the residual items e9 and e13 [80]. The specific results of the model modifications based on the MI values are shown in Table 3.

## 4.4 Model fitness results

After the model modification, the fitness of the model has been significantly improved. Moreover, the weight coefficients of each observed variable on its corresponding latent variable have passed the significance level test at 1%, indicating that the selected observed variables in this study effectively reflect the corresponding latent variables. The final fitness results are shown in Table 4, and Fig 3 presents the standardized path coefficients of the modified model.

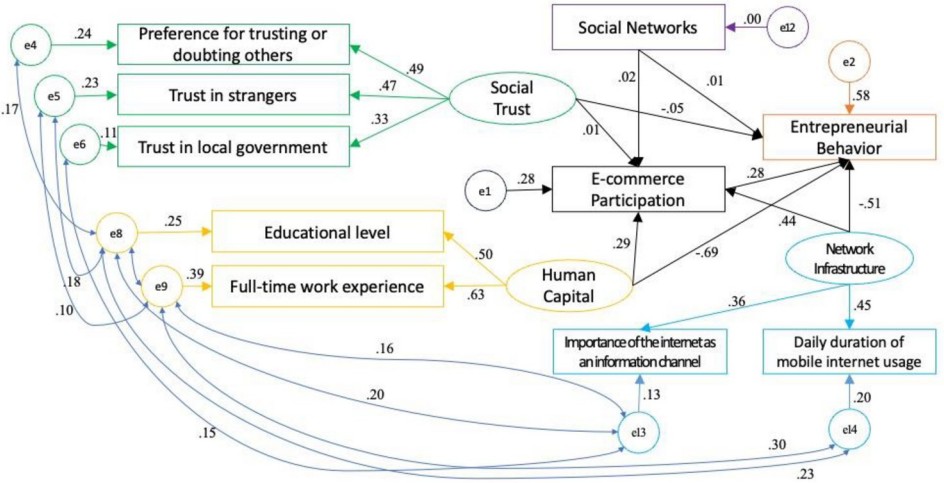

**Fig 3. Standardized path coefficients of the revised model.**

## 4.5 Path analysis

**4.5.1 Direct effects testing.**   According to Table 5, the standardized path coefficient of e-commerce participation on entrepreneurial behavior is positively significant. Thus, hypothesis H1 is supported. The standardized path coefficient is 0.282, indicating that e-commerce participation has a direct effect of 0.282 on farmers' entrepreneurial behavior. Holding other conditions constant, a one-unit increase in e-commerce participation can increase the probability of farmers' entrepreneurial choice by 28.2%. In other words, e-commerce participation contributes to an increased likelihood of farmer entrepreneurship to a certain extent. On the other hand, the standardized path coefficient of social networks on entrepreneurial behavior is 0.005, but it is not significant, indicating that social networks do not have a direct impact on farmers' entrepreneurial behavior. The standardized path coefficient of social trust on entrepreneurial behavior is -0.055, indicating that social trust has a direct and significant negative impact on farmers' entrepreneurial behavior. Considering relevant literature and questionnaire data, social trust can be divided into "special trust" based on kinship and geographical proximity, such as trust in parents, and "general trust" supported by non-kinship relationships, such as trust in strangers and government institutions [70]. People with higher levels of special trust tend to have higher risk aversion, thus reducing their probability of engaging in high-risk business activities, and vice versa [81]. Entrepreneurship is generally considered a high-risk activity. Looking at the selected observed variables under the latent variable "social trust," one of them is "liking or trusting others." This variable does not specifically refer to trust in parents or strangers. In practice, some respondents are likely to subconsciously answer based on trust in parents, relatives, and friends, i.e., "special trust." Holding other conditions constant, a higher level of special trust may inhibit farmers' entrepreneurial intentions and reduce their probability of entrepreneurship. Lastly, the standardized path coefficients of network infrastructure and human capital on entrepreneurial behavior are -0.509 and -0.691, respectively. This indicates that network infrastructure and human capital have significant direct negative effects on entrepreneurial behavior. The possibility of this outcome is that farmers with higher education levels and work experience possess higher human capital but may not necessarily have the necessary funds and other resources for entrepreneurship. In the absence of other necessary resources for entrepreneurship, education level alone cannot directly determine farmers' entrepreneurial behavior. Empirical studies show that farmers' experience of working outside their hometown inhibits their social evaluation and personal relationships locally, hindering their access to entrepreneurial resources through relatives and friends, and suppressing their entrepreneurial behavior [82].

**4.5.2 Indirect effect testing of e-commerce participation.**   To further analyze the indirect effect of e-commerce participation as a mediating variable on farmers' entrepreneurial

**Table 5. Coefficients and significance.**

| Path Description | Standard Error | Critical Ratio | Standardized Path Coefficient |
|---|---|---|---|
| E-commerce Participation → Entrepreneurial Behavior | 0.046 | 6.026 | 0.282*** |
| Social Network → Entrepreneurial Behavior | 0.000 | 0.374 | 0.005 |
| Social Trust → Entrepreneurial Behavior | 0.056 | -2.028 | -0.055** |
| Internet Infrastructure → Entrepreneurial Behavior | 0.001 | -7.017 | -0.509*** |
| Human Capital → Entrepreneurial Behavior | 0.068 | -15.915 | -0.691*** |

Note:

*, **, *** indicate statistical significance at the 10%, 5%, and 1% levels, respectively.

behavior, this study utilizes the Bootstrapping sampling method to examine the indirect effect generated by e-commerce participation as a mediating variable in the model [44]. The study employs AMOS software to perform Bootstrapping with 5000 repeated samples at a 95% probability level, in order to test the indirect effect of e-commerce participation as a mediating variable in the model. Fig 3 presents the standardized path coefficients of the modified model, and the results are shown in Table 6.

Table 6 represents that, at a 95% probability level, none of the three confidence intervals include zero. This means that, except for social networks, the empirical results indicate that e-commerce participation as a mediating variable for social trust, network infrastructure, and human capital has a significant effect on entrepreneurial behavior. Firstly, the indirect effect coefficient of e-commerce participation on the relationship between social trust and farmers' entrepreneurial behavior is 0.003. This implies that social trust has a significant positive impact on entrepreneurial behavior through e-commerce participation. In other words, holding other conditions constant, an increase of one unit in farmers' level of social trust would lead to a 0.3% increase in the probability of choosing entrepreneurship after engaging in e-commerce. This validates hypothesis H3. Secondly, e-commerce participation has significant positive effects on network infrastructure and human capital in relation to entrepreneurial behavior, with indirect effect coefficients of 0.124 and 0.082 respectively. This validates hypotheses H4 and H5. Holding all other conditions constant, an increase of one unit in network infrastructure and human capital would lead to a 12.4% and 8.2% increase, respectively, in the probability of farmers choosing entrepreneurial behavior after participating in e-commerce.

## 4.6 Discussion

First of all, the above empirical results show that rural e-commerce participation has a significant positive direct impact on farmers' entrepreneurial behavior, most of the previous studies pointed out that the level of development of rural e-commerce in the macro sense of the positive impact on the entrepreneurial behavior of farmers [55, 56, 83], while the study in this paper confirms the positive impact of individual farmers' e-commerce participation on farmers' entrepreneurial behavior, the conclusion for the government to formulate relevant policies to promote farmers' entrepreneurship provides new ideas. Entrepreneurship provides a new way of thinking, the existing literature on rural e-commerce to promote farmers' entrepreneurship policy recommendations are often to encourage villages or counties to actively apply for "e-commerce demonstration villages", e-commerce sales training for farmers, etc. [57], this kind of practice is indeed conducive to the promotion of rural e-commerce to a certain extent, the widespread dissemination of e-commerce, but the results are Not necessarily the best, such as scholars through case studies found that the government has developed Taobao users for e-commerce training, the effect is far greater than the farmers who have not been

**Table 6. Examination results of e-commerce participation as a mediating variable.**

| Indirect Effect Path | Indirect Effect Coefficient | 95% Confidence Interval | | Indirect Effect Effectiveness |
|---|---|---|---|---|
| Social Network → E-commerce Participation → Entrepreneurial Behavior | 0.006 | 0.000 | 0.001 | Not Supported |
| Social Trust → E-commerce Participation → Entrepreneurial Behavior | 0.003** | 0.003 | 0.018 | Supported |
| Internet Infrastructure → E-commerce Participation → Entrepreneurial Behavior | 0.124*** | 0.001 | 0.002 | Supported |
| Human Capital → E-commerce Participation → Entrepreneurial Behavior | 0.082*** | 0.088 | 0.186 | Supported |

Note:

*, **, *** indicate statistical significance at the 10%, 5%, and 1% levels, respectively.

involved in e-commerce, this part of the farmers tend to be more interested in e-commerce training and entrepreneurship, and for the mastery of the relevant skills faster [58], revealing that we should pay more attention to the importance of rural e-commerce participation. Secondly, e-commerce participation as a mediating variable has a significant positive effect on the relationship between human capital, social trust, network base and other variables respectively and farmers' entrepreneurial behavior, that is, the theoretical hypotheses of this paper are supported by the empirical results. As can be seen from Fig 3, the direct factors that are more affected are: "prefer to trust or doubt others" in the latent variable of social trust, "the length of time spent on the Internet on mobile devices" in the latent variable of network infrastructure, "full-time work experience" in the latent variable of human capital, and "the length of time spent on the Internet on mobile devices" in the latent variable of human capital. "full-time work experience", these factors are more likely to have a positive impact on farmers' entrepreneurial behavior under the effect of e-commerce participation, and this finding is to some extent conducive to providing preparatory ideas for farmers who have the intention to carry out entrepreneurial activities, and to improve the motivation and self-confidence of farmers' entrepreneurship.

Moreover, given that China was the world's largest developing country, the agricultural sector in China was dominated by small-scale family farmers and the country had a large number of mountainous regions, which made it difficult to mechanize agricultural production as well. E-commerce, however, bridges the gap between Chinese farmers and external producers, promotes the development of the local agricultural economy, provides farmers with diversified distribution channels, effectively solves the information asymmetry between farmers and final consumers faced in the process of farmers' entrepreneurship, and improves the profit margins of farmers as well as the efficiency of agricultural product distribution [61, 75, 84]. The most direct impact of rural e-commerce participation on farmers' agricultural entrepreneurial behavior is that it can make the farmers of agricultural products network sales of low-cost, low-threshold, and the possibility of understanding the market mechanism is greater. The traditional sales channels for agricultural products rely on local markets, but rural e-commerce participation helps farmers to directly reach a broader market. In addition, rural e-commerce participation is conducive to improving the efficiency of farmers' agricultural production, such as more efficiently purchasing good and inexpensive agricultural production materials and agricultural machinery, etc., improving productivity and accumulating start-up capital. Not only that, rural e-commerce participation promotes the modernization and refined management of agriculture with its unique advantages. Through the network e-commerce platform, farmers can obtain the latest information on agricultural technology and management methods, carry out refined management of agricultural production, and promote the modernization of agriculture. In this study, to analysis the impact of e-commerce participation on agricultural entrepreneurship, we have chosen a sample of 4,057 farmers, 1,700 samples with entrepreneurial behavior. The concept of "agricultural entrepreneurship behavior" was used to measure the extent to which farmers engage in entrepreneurial activities in the agricultural field. A value of 1 was assigned if the entrepreneurial activity was related to agriculture, and 0 if it was unrelated to agriculture. It was found that 1,270 farmers (74.71% of the sample) were engaged in agricultural entrepreneurship behavior. Using AMOS software, a revised conceptual model was employed, and the entrepreneurial behavior variable was replaced with the agricultural entrepreneurship behavior variable for analysis. The standardized path coefficients are shown in Fig 4, and the results are presented in Table 7.

Table 7 indicates that several key findings emerge. Firstly, the standardized path coefficient of e-commerce participation on agricultural entrepreneurship behavior is 0.296, and the direct effect is significant. This indicates that e-commerce participation has a significant positive

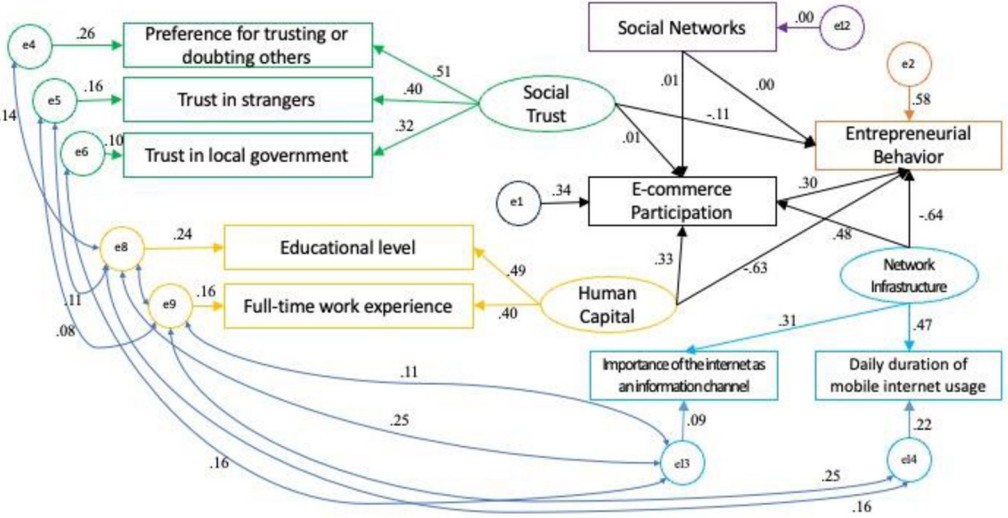

**Fig 4. Standardized path coefficients diagram.**

impact on agricultural entrepreneurship. Holding other factors constant, an increase of one unit in e-commerce participation level leads to a 29.6% increase in the probability of farmers engaging in agricultural entrepreneurship. Secondly, the indirect effects of e-commerce participation on social networks and social trust in relation to agricultural entrepreneurship are not significant. This implies that e-commerce participation does not have an indirect effect through these mechanisms. Lastly, e-commerce participation has significant and positive indirect effects on the relationships between network infrastructure and human capital with agricultural entrepreneurship. Notably, the empirical results show that these two variables have a significant negative direct effect on agricultural entrepreneurship. In other words, under unchanged conditions, higher levels of network infrastructure and human capital tend to decrease the probability of farmers engaging in agricultural entrepreneurship. However, when e-commerce participation is introduced as a mediating variable, it increases the likelihood of farmers with higher levels of network infrastructure and human capital choosing agricultural entrepreneurship. Considering the empirical results and the real-world perspective, this situation is likely due to the fact that many farmers initially engage in agricultural activities. The reason for this situation may be that farmers with higher levels of human capital have more

**Table 7. Standardized path coefficients and significance.**

| Direct Effect Path | Standard Error | Critical Ratio | Standardized Path Coefficient | |
|---|---|---|---|---|
| E-commerce Participation → Agricultural Entrepreneurship | 0.094 | 2.728 | 0.296** | |
| **Indirect Effect Path** | **Indirect Effect Coefficient** | **95% Confidence Interval** | **Indirect Effect Effectiveness** | |
| Social Network → E-commerce Participation → Agricultural Entrepreneurship | -0.003 | -0.001 | 0.000 | Not Supported |
| Social Trust → E-commerce Participation → Agricultural Entrepreneurship | 0.003 | -0.033 | 0.044 | Not Supported |
| Internet Infrastructure → E-commerce Participation → Agricultural Entrepreneurship | 0.144*** | 0.001 | 0.002 | Supported |
| Human Capital → E-commerce Participation → AgriculturalEntrepreneurship | 0.099*** | 0.101 | 0.477 | Supported |

employment options, and due to the influence of traditional Chinese culture, most farmers have a stronger sense of risk aversion, and some of them are more inclined to choose less risky and stable jobs without having been exposed to e-commerce, which is a low-cost entrepreneurial activity [62]. Farmers with higher levels of human capital and network infrastructure possess the ability to gather and process information. With e-commerce participation, these farmers gain a better understanding of market mechanisms. Additionally, considering their existing skills and resources, they are more likely to choose agricultural entrepreneurship activities that involve selling agricultural products through online platforms. These types of activities are relatively easier to start and have lower costs, increasing the probability of engagement in agricultural entrepreneurship.

## 5. Conclusions and policy recommendations

This study utilized the 2020 CFPS data and employed SEM model to comprehensively analyze the direct impact of e-commerce participation on entrepreneurial behavior among farmers under the influence of various factors such as social networks, social trust, internet infrastructure, and human capital. The study also explored the indirect effects and mechanisms of e-commerce participation as a mediating variable, further analyzing the impact and mechanisms of e-commerce participation on agricultural entrepreneurial behavior. The research reveals that: first, e-commerce participation significantly promotes entrepreneurial behavior among farmers; second, social trust, internet infrastructure, and human capital, under the mediating role of e-commerce participation, facilitate entrepreneurial behavior among farmers; third, e-commerce participation has a positive and significant impact on agricultural entrepreneurial behavior. Farmers with higher levels of internet infrastructure and human capital are more likely to choose agricultural entrepreneurship under the influence of e-commerce participation. Given these revelations, the following policy recommendations are proposed.

First, government or authorities should continue to improve and accelerate the construction of cold chain logistics and internet infrastructure for rural e-commerce. Due to factors such as rural economic development and convenience, many people are unwilling to stay in rural areas. The rapid development of cold chain logistics is conducive to improving the entrepreneurial environment in rural areas and promoting farmers' choice of e-commerce entrepreneurship. According to questionnaire data, the largest proportion of entrepreneurial activities among farmers is in their own agricultural production and operation, followed by individual businesses and private enterprises. Among the farmers who engage in entrepreneurial activities, approximately 74.71% are engaged in agricultural entrepreneurship. Particularly with the development of rural e-commerce, the sale and operation of fresh agricultural products have become an important source of income for farmers' individual businesses or private enterprises. Fresh agricultural products are perishable and susceptible to damage, and cold chain logistics directly affect the speed and quality of their transportation. Therefore, it is necessary to accelerate the construction of rural logistics infrastructure, especially cold chain transportation, to enhance the entrepreneurial confidence of farmers choosing agricultural entrepreneurship. At the same time, the government should increase efforts to popularize the internet in rural areas and improve internet coverage, laying the foundation for creating a favorable e-commerce environment. The internet can help farmers quickly obtain information about e-commerce entrepreneurship from a vast amount of information. Many farmers are skeptical and cautious about new things, and their fundamental reason for not participating in e-commerce activities lies in their lack of understanding. The internet can help farmers acquire various knowledge about e-commerce and understand the basic transaction rules, enabling them to establish a good market concept, strengthen their connections with the outside world,

enhance their market integration capabilities, and increase their confidence in choosing entrepreneurship. Furthermore, the internet can help farmers establish a richer social network, enhance their social capital, and promote their entrepreneurial choices.

Second, local governments should actively promote policies and projects related to e-commerce in rural areas, creating a favorable environment for e-commerce participation among farmers and enhancing their trust in local governments. As indicated by the research conclusions, farmers with higher levels of trust in the government are more likely to choose entrepreneurship through e-commerce participation. Combined with the questionnaire data, approximately 30% of the sampled farmers have a moderate level of trust in the government, and around 40% of farmers rate their trust in the government above 5 points. This indicates that most of the sampled farmers have a relatively high level of trust in the government, reflecting the good achievements of grassroots governance in China and the superiority of the basic political system. However, there are still some farmers who do not have a high level of trust in the government. Therefore, it is necessary to continue to improve governance levels and promote government trust by creating a favorable environment for e-commerce participation. Firstly, various media platforms should vigorously promote the benefits that e-commerce participation can bring to farmers, such as the affordability of online products. Secondly, young people should be encouraged to lead the participation of older people in e-commerce, bridging the generational gap in e-commerce participation. Thirdly, e-commerce rights protection institutions should be established locally to help farmers effectively address technical difficulties they may encounter in e-commerce participation and protect their legitimate rights and interests, alleviating their concerns about security risks in e-commerce participation. As a result, farmers will be more willing to participate in e-commerce activities, accumulate prior experience in e-commerce participation, and promote entrepreneurship among farmers.

Finally, governments should continue to improve the education level in rural areas and enhance the entrepreneurial environment for farmers engaged in e-commerce. The research conclusions indicate that farmers with higher education levels and more full-time work experience are more likely to choose entrepreneurship under the influence of e-commerce participation. Observing the sample data, 30% of the interviewed farmers have primary school education or below, 40% have junior high school education, and only around 10% have college education or above. It shows that the education level of most farmers is relatively low. A low level of education will result in a lower level of awareness and inadequate skills related to e-commerce participation, which will affect farmers' access to prior experience in the e-commerce industry. To some extent, it will restrict farmers' ability to obtain relevant entrepreneurial information and hinder their ability to apply the entrepreneurial knowledge they have learned. In some rural areas, especially underdeveloped and remote mountainous regions, there are still many shortcomings in education. Therefore, it is necessary to continuously improve the education level in rural areas through adult education and vocational training. In underdeveloped counties, townships, and villages, the construction of high schools and vocational schools should be newly established or expanded. Efforts should be made to strengthen the construction of agricultural colleges, agricultural vocational colleges, and agricultural disciplines. At the same time, it is important to pay attention to the employment and entrepreneurship environment for farmers engaged in e-commerce. The government should increase efforts to popularize knowledge about e-commerce entrepreneurship, national policies, and local support policies. It should also provide relevant skills and vocational training based on the needs of farmers' e-commerce employment and entrepreneurship. Simplifying the procedures for farmers' e-commerce entrepreneurship qualification reviews, increasing financial support for e-commerce entrepreneurship, regulating relevant laws and regulations, guiding local farmers to legally utilize e-commerce for employment and entrepreneurship are also

crucial. Meanwhile, relevant employment policies should be introduced to strongly support the development of local e-commerce and logistics enterprises, drive rural employment and entrepreneurship, help farmers accumulate industry experience and entrepreneurial capital.

## 5.1 Limitation and further research direction

It is important to acknowledge that the this study has certain limitations. Due to limitations in data availability, the selection of indicators for assessing rural e-commerce participation is constrained, leading to a narrow focus on a single dimension. Currently, research predominantly examines the influence of rural e-commerce participation on entrepreneurial behavior and agricultural entrepreneurship. However, it is essential to extend these investigations to encompass a broader perspective, including an exploration of the impact of rural e-commerce participation on the rural entrepreneurial process and entrepreneurial performance. In addition, understanding the influence of rural e-commerce participation on the rural entrepreneurial process represents a crucial research area. In-depth analyses of how rural e-commerce participation inspires, influences decision-making, and shapes the execution of entrepreneurial activities among farmers can provide valuable insights into the challenges and opportunities they encounter throughout their entrepreneurial Process. Such investigations facilitate an examination of the mechanisms through which rural e-commerce fosters entrepreneurial engagement among farmers, thereby shedding light on the decision-making and actions taken by farmers during the entrepreneurial process. Moreover, investigating the impact of rural e-commerce participation on farmers' entrepreneurial performance is of significant scholarly interest. This research direction involves an examination of the effects of rural e-commerce participation on various aspects of entrepreneurial ventures, such as market performance, profitability, and sustainable development. Through comprehensive evaluations of the impact of rural e-commerce on farmers' entrepreneurial projects, researchers can gain a deeper understanding of their market performance, profitability, and long-term viability. By examining the multifaceted effects of rural e-commerce participation, scholars can provide comprehensive insights into the dynamics between rural e-commerce and farmers' entrepreneurial activities, thereby informing policy decisions and facilitating the development of effective strategies to promote rural entrepreneurship.

## Supporting information

**S1 Data.**
(XLSX)

## Author Contributions

**Conceptualization:** Huayuan Wu.

**Data curation:** Huayuan Wu, Tianqi Zhu.

**Formal analysis:** Haiying Lin, Tianqi Zhu.

**Investigation:** Haonan Chen.

**Methodology:** Tianqi Zhu.

**Project administration:** Wenlong Li.

**Resources:** Haihua Lin, Haonan Chen.

**Software:** Haihua Lin.

**Supervision:** Wenlong Li.

**Visualization:** Haihua Lin.

**Writing – original draft:** Haiying Lin.

**Writing – review & editing:** Haiying Lin, Muhammad Umer Arshad.

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
