## [Decision Letter · Decision Letter 0]

20 Dec 2023

PONE-D-23-33644The impact of rural e-commerce participation on farmers' entrepreneurial behavior: Evidence based on CFPS data in ChinaPLOS ONE

Dear Dr. Li,

Thank you for submitting your manuscript to PLOS ONE. After careful consideration, we feel that it has merit but does not fully meet PLOS ONE’s publication criteria as it currently stands. Therefore, we invite you to submit a revised version of the manuscript that addresses the points raised during the review process.

Please submit your revised manuscript by Feb 03 2024 11:59PM. If you will need more time than this to complete your revisions, please reply to this message or contact the journal office at plosone@plos.org. Please include the following items when submitting your revised manuscript:A rebuttal letter that responds to each point raised by the academic editor and reviewer(s). You should upload this letter as a separate file labeled 'Response to Reviewers'.A marked-up copy of your manuscript that highlights changes made to the original version. You should upload this as a separate file labeled 'Revised Manuscript with Track Changes'.An unmarked version of your revised paper without tracked changes. You should upload this as a separate file labeled 'Manuscript'.

We look forward to receiving your revised manuscript.

Kind regards,

Ioana Gutu, Postdoctoral

Academic Editor

PLOS ONE

3. PLOS requires an ORCID iD for the corresponding author in Editorial Manager on papers submitted after December 6th, 2016. Please ensure that you have an ORCID iD and that it is validated in Editorial Manager. To do this, go to ‘Update my Information’ (in the upper left-hand corner of the main menu), and click on the Fetch/Validate link next to the ORCID field. This will take you to the ORCID site and allow you to create a new iD or authenticate a pre-existing iD in Editorial Manager. Please see the following video for instructions on linking an ORCID iD to your Editorial Manager account: https://www.youtube.com/watch?v=_xcclfuvtxQ.

Additional Editor Comments:

Dear Authors,

Thank you for submitting your manuscript to PLOS ONE. After careful consideration, we have decided that in order to meet our criteria for publication, you need to improve your manuscript.

Specifically, I think the project as a whole has potential merit, but needs much more serious thought and consideration. To continue with the project, I would urge the Authors to English proof the entire manuscript, starting withe the Abstract section. Moreover, clarifications in regard to the statistical analysis considered thresholds as for preventing Results from being biased are necessary. 

The manuscript is not adequately referenced.

Please revise the manuscript according to the referees' comments and upload the revised file.

Best regards,

Reviewers' comments:

Reviewer's Responses to Questions

**Comments to the Author**

1. Is the manuscript technically sound, and do the data support the conclusions?

Reviewer #1: Yes

Reviewer #2: Yes

2. Has the statistical analysis been performed appropriately and rigorously? 

Reviewer #1: Yes

Reviewer #2: Yes

3. Have the authors made all data underlying the findings in their manuscript fully available?

Reviewer #1: Yes

Reviewer #2: Yes

4. Is the manuscript presented in an intelligible fashion and written in standard English?

Reviewer #1: Yes

Reviewer #2: Yes

5. Review Comments to the Author

Reviewer #1: Remarks/ questions regarding the paper with the title "The impact of rural e-commerce participation on farmers' entrepreneurial behavior: Evidence based on CFPS data in China" [an interesting research idea] - in order to improve paper quality:

A. general remarks/ questions:

- contribution should be (more) clarified. The authors should further clarify what the contribution of the paper is, what is new in this paper? Why should it be published? What is the literature gap covered by this paper? (must be improved) What is the associated interest of this contribution? Has anyone previously suggested the need and interest in developing this specific contribution?

- the literature (review) approached in the manuscript should serve to synthesize the state of the art in the topic addressed, to describe the main specific contributions made to date, what is the gap that the work tries to fill, how the previous contributions relate to the contribution that is intended to be made in this paper and, if it is the case, who previously suggested the need to make the analysis included in this new study.

B. particular remarks/ questions:

- paper abstract must be clear;

- chapter 2 of the manuscript is relatively limited -> considering the fact that 4 hypotheses are issued based on it; must be substantial improved;

- regarding "CFPS database" it would be appropriate to present an associated link;

- on 3.3 sub-chapter the authors mention "... practical considerations and referencing previous literature [64-66], a total of

10 observed variables are selected ..."; therefore explain what it means "practical considerations" and the mentioned references [64-66] are not relatively few/ limited? Also why (at least or at most) "five latent variables" ("entrepreneurial behavior, e-commerce participation, human capital, network infrastructure and social networks")?

- [on 4.1 paper sub-chapter] authors do not consider that values associated to Cronbach's alpha (0.642) and KMO (0.797) relatively weak/ medium?

- "4.6 Discussion" section are limited and insufficient; a correlation with literature are needed; please improve substantial;

and so on.

Reviewer #2: A very good article with all the necessary elements. Written to a very high level of content. Good literature review, high level of statistical analysis with correct conclusions. Properly indicated directions for future research and limitations of the empirical studies carried out.

6. PLOS authors have the option to publish the peer review history of their article (what does this mean?). If published, this will include your full peer review and any attached files.

Reviewer #1: No

Reviewer #2: No

---

## [Author Response · Author response to Decision Letter 0]

11 Feb 2024

Response to Reviewer 1 Comments

Dear Reviewer,

Thank you very much for your so professional and so detailed comments and guidance, it is of great help to improve our manuscript. Thank you very much for your comments of the contribution on this paper, it has greatly encouraged our research enthusiasm. According to your guidance, we have improved the revised manuscript point by point. And your comments were responded and explained in detail as follows.

General Comment

Point 1: Contribution should be (more) clarified. The authors should further clarify what the contribution of the paper is, what is new in this paper? Why should it be published? What is the literature gap covered by this paper? (must be improved) What is the associated interest of this contribution? Has anyone previously suggested the need and interest in developing this specific contribution?

Response 1: Thank you very much for the expert's valuable advice, As the experts have pointed out, the contribution of should be (more) clarified.The authors have thought deeply about this and have systematised the contribution of this paper in the second part of the article. Overall, the contribution of this paper is in the following two parts:

1. Theoretical Contributions

(1) Most of the existing literature involves farmers in individual sample areas, while the sample object selected in this paper is farmers across the country, which is relatively more general. 

(2) Most of the previous studies explore the level of e-commerce in inter-provincial or county areas, while this paper expands the empirical study of rural e-commerce on farmers' entrepreneurial behaviour and agriculture-related entrepreneurial behaviour from a micro perspective, with the research focusing on the impact of individual farmers' e-commerce activities on farmers' entrepreneurship. This study focuses on the impact of individual farmers' e-commerce activities on farmers' entrepreneurship, combines the existing authoritative literature and the real-world background, analyses the theoretical mechanism of the impact of rural e-commerce participation on farmers' entrepreneurial behaviour and agriculture-related entrepreneurial behaviour under the role of multiple factors, and explores the key influencing factors, which is conducive to supplementing and perfecting the theoretical system of farmers' entrepreneurial behaviour, especially the theory of agriculture-related entrepreneurial behaviour. 

(3) The importance of rural e-commerce participation on agricultural entrepreneurial behaviour is seldom mentioned in the existing literature, especially in developing countries such as China, which has a large number of mountainous areas and where mechanisation of agricultural production is difficult. E-commerce is a bridge between Chinese farmers and external producers, and the most direct impact of rural e-commerce participation on farmers' agricultural entrepreneurial behaviour is that it can make it possible for farmers to have a greater understanding of the low-cost, low-threshold, and market mechanism of agricultural product online sales, which is not negligible in terms of the importance of rural e-commerce for farmers who have long been making their living through agricultural production. There is no literature to explore this phenomenon in depth, this paper to some extent enriches the theoretical research in the field of rural e-commerce participation.

2. Practical Contributions

(1) This study uses research data to provide an in-depth discussion on the importance of farmers' participation in e-commerce and in entrepreneurial activities, and tests the intrinsic factors affecting farmers' entrepreneurship in empirical analyses, such as human capital and social capital, which can provide theoretical references for farmers who hope to achieve income increase and improve their living standards through entrepreneurship, so that they can have certain ideas to prepare for entrepreneurship and improve farmers' entrepreneurial motivation and self-confidence. (2) The research results of this paper can help the relevant government departments to formulate targeted policies to promote farmers' entrepreneurship and agricultural development, create a good entrepreneurial environment, and improve the level of grass-roots governance to provide a certain micro reference basis.

See specifically: Line 211-265. 

Point 2: The literature (review) approached in the manuscript should serve to synthesize the state of the art in the topic addressed, to describe the main specific contributions made to date, what is the gap that the work tries to fill, how the previous contributions relate to the contribution that is intended to be made in this paper and, if it is the case, who previously suggested the need to make the analysis included in this new study.

Response 2: The valuable comments of the experts are much appreciated. As pointed out by the experts, the literature (review) in the manuscript should help to synthesise the recent advances in the subject covered and describe the main specific contributions made so far. The authors believe that your suggestion is very pertinent, therefore, in the second part of the paper, the authors have conducted a systematic review of the existing literature in the related field, and the authors believe that the previous studies have laid a good theoretical foundation for the present study, and the marginal contribution of the present study to the field is mainly in the following ways: expanding the empirical research on rural e-commerce on farmers' entrepreneurial behaviour as well as the micro-views related to the farmers' agro-entrepreneurship behaviours, and the research focuses on the impact of individual farmers' e-commerce activities on farmers' entrepreneurship. In addition, studies by scholars such as Pindado (2017) and Seuneke (2013) mention that a better understanding of the entrepreneurial behaviour of the agricultural sector can be an essential tool for promoting the vitality and competitiveness of rural areas, but agricultural entrepreneurship involves a specific contact with the rural natural environment, which tends to make the agro-entrepreneurs face some specific challenges, emphasising the study of the agricultural sector's entrepreneurial The need to study the behaviour of entrepreneurship in the agricultural sector, especially in future research should be more focused on understanding how agricultural entrepreneurs acquire entrepreneurial competencies, and the content of this paper provides some ideas for answering this question.

See specifically: Line 111-209.

Special Comment

Point 1: Paper abstract must be clear.

Response 1: Thank you very much for the expert's careful suggestions. According to your suggestions, the authors, after careful consideration, have adjusted the abstract part of this paper, and the adjusted abstract mainly describes the research background, research content, conclusions, policy recommendations and main innovations of this paper.

See specifically: Line 16-45 .

Point 2: Chapter 2 of the manuscript is relatively limited → considering the fact that 4 hypotheses are issued based on it; must be substantial improved.

Response 2: Thank you very much for the expert's careful suggestions. After careful consideration, the authors further improved the theoretical analysis of the four hypotheses and more specifically elaborated the mechanism of direct and indirect influence of rural e-commerce participation on farmers' entrepreneurial behavior.

See specifically: Line 287-423.

Point 3: Regarding "CFPS database" it would be appropriate to present an associated link;

Response 3: Thank you very much for the expert's careful suggestions. This is the direct link to the CFPS database: http://www.isss.pku.edu.cn/cfps/download

Point 4: On 3.3 sub-chapter the authors mention "... practical considerations and referencing previous literature [64-66], a total of 10 observed variables are selected ..."; therefore explain what it means "practical considerations" and the mentioned references [64-66] are not relatively few/ limited? Also why (at least or at most) "five latent variables" ("entrepreneurial behavior, e-commerce participation, human capital, network infrastructure and social networks")? 

Response 4: Thank you very much for the expert's careful suggestions. As you said, there are relatively few references here, and the authors have added four after careful consideration and reviewing the literature, and adjusted the references accordingly. In addition, "practical considerations" refers to the ideas put forward by the authors in consideration of actual cases that happened in real life, such as China has a special term "Taobao Village", according to the in-depth investigation of journalists and scholars, many farmers in these villages are involved in e-commerce activities and the idea of entrepreneurship, and ultimately succeed in entrepreneurship! Farmers have some obvious qualities, such as they tend to be well-liked, have a certain level of education, and have the experience of going out to work, etc. These cases do not only happen to a small number of people, but most scholars in the existing literature tend to consider the influence of only one factor, and the results of the study appear to be diametrically opposed to each other, for example, Zhou Zixin (2021) pointed out that the higher the level of education, the lower the probability of farmers' entrepreneurship, but the lower the probability of entrepreneurship, the lower the probability of entrepreneurship. , the lower the probability of farmers' entrepreneurship, but the empirical study of Xingwang et al. (2018) pointed out that the level of education is conducive to promoting farmers' entrepreneurial behavior. This triggered the authors' thinking, we believe that the level of education or social capital and other factors may not be a direct determinant of entrepreneurial behavior, considering that e-commerce participation is conducive to farmers' understanding of low-cost entrepreneurial methods, and farmers' participation in e-commerce inevitably needs to have a certain level of knowledge, so when farmers with a certain level of knowledge recognize a low-cost and relatively low-risk entrepreneurial methods through this pathway Then, when farmers with a certain level of knowledge recognize a low-cost and relatively low-risk way of entrepreneurship through this way, will they be more willing to choose entrepreneurship? Therefore, the authors further reviewed a large amount of literature, summarized the main factors affecting farmers' entrepreneurial behavior, and then selected five potential variables in combination with actual cases, of which "entrepreneurial behavior" and "e-commerce participation" are the two key variables in this paper, while "human capital" and "e-commerce participation" are the two key variables in this paper. "Human capital", "social capital" and "network base" are the three latent variables that the authors believe can influence entrepreneurial behavior through the mediating variable of e-commerce participation. In addition, structural equation modeling can well represent the influence mechanism of this study, which is located in Chapter 2 of this paper.

See specifically: Line 287-423.

Point 5: [on 4.1 paper sub-chapter] authors do not consider that values associated to Cronbach's alpha (0.642) and KMO (0.797) relatively weak/ medium?

Response 5: Thank you very much for the expert's careful suggestions. In response to your question, the authors would like to explain to you that this paper uses secondary data from the CFPS database, which is a national panel survey data hosted by the Institute of Economics of the Chinese Academy of Social Sciences (CASS), and is mainly used to study the socio-economic status of China's residents, and its reliability and validity of the data have been recognized by a wide range of scholars. However, since the original data are not collected for the specific research questions in this paper, it will affect the Cronbach's alpha and KMO values to a certain extent. However, as shown by the empirical analysis of structural equation modeling in the paper, the results of the study are significant and some of them are consistent with the results of previous studies, so the authors believe that these two coefficients are applicable to the study of this paper.

Point 6: "4.6 Discussion" section are limited and insufficient; a correlation with literature are needed; please improve substantial.

Response 6: Thank you very much for the expert's careful suggestions. The authors have refined the specifics of the "4.6 Discussion" section and added relevant literature support.

See specifically: Line 723-772 .

---

## [Decision Letter · Decision Letter 1]

27 Feb 2024

The impact of rural e-commerce participation on farmers' entrepreneurial behavior: Evidence based on CFPS data in China

PONE-D-23-33644R1

Dear Dr. Wenlong Li,

We’re pleased to inform you that your manuscript has been judged scientifically suitable for publication and will be formally accepted for publication once it meets all outstanding technical requirements.

Kind regards,

Ioana Gutu, Postdoctoral

Academic Editor

PLOS ONE

Additional Editor Comments (optional):

Reviewers' comments:

Reviewer's Responses to Questions

**Comments to the Author**

1. If the authors have adequately addressed your comments raised in a previous round of review and you feel that this manuscript is now acceptable for publication, you may indicate that here to bypass the “Comments to the Author” section, enter your conflict of interest statement in the “Confidential to Editor” section, and submit your "Accept" recommendation.

Reviewer #1: All comments have been addressed

Reviewer #2: All comments have been addressed

2. Is the manuscript technically sound, and do the data support the conclusions?

Reviewer #1: Yes

Reviewer #2: Yes

3. Has the statistical analysis been performed appropriately and rigorously? 

Reviewer #1: Yes

Reviewer #2: Yes

4. Have the authors made all data underlying the findings in their manuscript fully available?

Reviewer #1: Yes

Reviewer #2: Yes

5. Is the manuscript presented in an intelligible fashion and written in standard English?

Reviewer #1: Yes

Reviewer #2: Yes

6. Review Comments to the Author

Reviewer #1: I have no other recommendations associated with the article!

.....................................................................................

Reviewer #2: Article at a very high level of content, both in the theoretical and practical parts. Statistical analysis at a very advanced level.

7. PLOS authors have the option to publish the peer review history of their article (what does this mean?). If published, this will include your full peer review and any attached files.

Reviewer #1: No

Reviewer #2: No

---

## [Editor Report · Acceptance letter]

14 Mar 2024

PONE-D-23-33644R1 

PLOS ONE

Dear Dr. Li, 

I'm pleased to inform you that your manuscript has been deemed suitable for publication in PLOS ONE. Congratulations! Your manuscript is now being handed over to our production team.

Kind regards, 

on behalf of

Dr. Ioana Gutu 

Academic Editor

PLOS ONE